# Association Between Protein-Rich Foods, Nutritional Supplements, and Age of Natural Menopause and Its Symptoms

**DOI:** 10.3390/nu17020356

**Published:** 2025-01-20

**Authors:** Yilin Yang, Yehuan Yang, Zhenghua Yong, Li Yang, Yanxia Zhao, Mengke Yan, Ruimin Zheng, Xiaomin Luo

**Affiliations:** 1Chinese Center for Disease Control and Prevention, Beijing 100000, China; yyl8381@163.com (Y.Y.); yzh990603@163.com (Z.Y.); yanmengke1223@163.com (M.Y.); 2National Center for Women and Children’s Health, National Health Commission of the People’s Republic of China, Beijing 100000, China; yangyehuan@ncwchnhc.org.cn (Y.Y.); yangli@ncwchnhc.org.cn (L.Y.); zhaoyanxia@ncwchnhc.org.cn (Y.Z.)

**Keywords:** natural menopause age, menopausal symptoms, fish, milk, soy

## Abstract

Objective: To investigate the relationship between protein-rich foods, various nutritional supplements, and age of natural menopause and its symptoms. Methods: This study was a large-scale cross-sectional survey. A multi-stage stratified random sampling method was used to select a sample of 52,347 residents aged 35–60 years from 26 districts/counties across 13 cities in 12 provinces in China. The mean natural menopause age was represented by the average and median, and logistic regression models were employed to examine the relationship between the intake of protein-rich foods, use of nutritional supplements, and natural menopause age as well as menopausal symptoms. Results: The average natural menopause age of the study population was 49.46 (±3.22) years, with a median age of 50 years. Logistic regression analysis revealed that with increasing frequency of fish consumption, the natural menopause age was delayed and the severity of menopausal symptoms gradually decreased, especially in relation to physical symptoms, psychological symptoms, and urogenital symptoms, which showed significant negative correlations. Milk and soy products were significantly negatively correlated with various dimensions of menopausal symptoms. Calcium and iron supplements were significantly positively correlated with the severity of menopause symptoms. Subgroup analysis by different age groups and premenopausal versus postmenopausal status showed minimal impact on the study results. In the population with BMI ≥ 18.5, fish consumption was significantly negatively correlated with menopausal symptoms. In the population with BMI between 18.5 and 27.9, milk consumption was significantly negatively correlated with menopausal symptoms. In all the populations, soy product consumption was significantly negatively correlated with menopausal symptoms. Conclusions: The intake of fish, milk, and soy products plays a role in alleviating the severity of menopausal symptoms, particularly in delaying natural menopause, with the effect of fish intake being especially significant. Calcium and iron supplements may play a role in exacerbating the severity of menopausal symptoms.

## 1. Introduction

Natural menopause is defined as the permanent cessation of menstruation resulting from the irreversible loss of ovarian follicular function. It is diagnosed retrospectively after 12 consecutive months of amenorrhea without any pathological or physiological cause, in which the final menstrual period is considered to represent natural menopause [1]. Menopause marks the end of the female reproductive cycle, accompanied by ovarian function decline and cessation of fertility. This results in hormonal changes in the body, leading to the onset of a range of symptoms. The timing of menopause and the severity of the related symptoms may be influenced by various factors, including genetic factors, smoking and alcohol consumption, dietary habits, medical history, current health status, obesity, and socioeconomic status [2,3,4,5,6]. In recent years, research on diet and menopause has been increasing. Studies from abroad have found that protein-rich foods (such as soy and dairy products) and a low-fat, high-protein Mediterranean diet can alleviate menopausal symptoms [7]. Similarly, research indicates that a diet rich in n-3 polyunsaturated fatty acids and an appropriate balance of n-6 polyunsaturated fatty acids can help alleviate menopausal symptoms [8,9,10]. Some studies suggest that calcium supplements can improve osteoporosis in menopausal women [11], while excessive iron intake can also affect osteoporosis [12]. However, the intake of solid fats, snacks, and red meat has been shown to worsen menopausal symptoms [13]. Additionally, the intake of protein-rich foods, vitamin D, calcium supplements, and dairy products can delay the onset of menopause [14,15], and a vegetarian diet may advance the age of natural menopause [16]. There is a lack of research in China on the effects of protein-rich foods and nutritional supplements on the age of natural menopause and menopausal symptoms. Given the significant differences in dietary habits between China and other countries, it is challenging to directly apply foreign research findings. Therefore, it is necessary to conduct studies on the impact of protein-rich foods and nutritional supplements on menopause age and menopausal symptoms among Chinese women, exploring the potential role of dietary habits in women’s reproductive health. This study aims to analyze the relationship between the intake of protein-rich foods and nutritional supplements and the age of natural menopause and menopausal symptoms, providing scientific evidence and data support for dietary guidance and menopause health management for perimenopausal women in China.

## 2. Materials and Methods

### 2.1. Study Design

A large-scale cross-sectional survey design was used, combined with a multi-stage stratified cluster sampling method. A total of 26 districts and counties were selected from 12 provinces and 13 cities across Eastern, Central, and Western China as the survey locations. The sample sizes for each region were allocated according to the population proportions by province and by age group of women, based on the data from the 2021 Census of China released on 11 May 2021. Data were collected at a single time point using structured questionnaires to assess factors associated with menopause age and its symptoms. The sample size was calculated using the cross-sectional survey formula.

This study was approved by the Ethical Review Committee of the National Center for Women and Children’s Health, Chinese Center for Disease Control and Prevention (Approval No: FY2024-03, 5 March 2024). This study was funded by the National Key R&D Program of China.

#### Inclusion and Exclusion Criteria

The inclusion criteria for participants were non-pregnant women aged 35–60 years, with local residency or having lived in the area for more than 6 months (55,056 participants). Exclusion criteria were incomplete information (2253 participants) and a history of premenopausal hysterectomy (138 participants), resulting in a total of 52,665 data entries, with an effective response rate of 95.6%. Women with non-natural menopause (301 participants) and a history of bilateral adnexectomy (17 participants) were also excluded, leaving 52,347 participants for final analysis. All the participants were required to sign an informed consent form and voluntarily complete all survey content (Figure 1).

### 2.2. Questionnaire Survey

All the participants completed an electronic questionnaire and physical examination under the guidance of medical staff. The primary content of the survey included demographic information, lifestyle factors, physical examination, menopause status (whether menopause, age at menopause, last menstruation time, and menopause causes), menstrual history, and an assessment of menopausal symptoms using the modified Kupperman index.

### 2.3. Statistical Analysis

Data analysis was performed using Stata 17.0, using the Kolmogorov–Smirnov test to assess whether the data met the normality assumption. Normally distributed continuous variables were presented as mean ± standard deviation (±s), while non-normally distributed continuous variables were presented as medians and interquartile range (IQR). Categorical data were presented as percentages. The natural age of menopause was calculated based on the time of the last menstruation and the participant’s birthdate. Dietary data were collected from the survey questionnaire. Food intake frequency was categorized as follows: 1 = “Never”, 2 = “≤2 times/week”, 3 = “3–5 times/week”, and 4 = “≥6 times/week”. Nutrient supplement use was a binary variable: 1 = “Not used”, 2 = “Used”. Menopause age was divided into five categories (<40, 40–44, 45–49, 50–54, 55–60). BMI was categorized as follows: underweight (<18.5), normal (18.5–23.9), overweight (24.0–27.9), and obese (≥28.0). Age was divided into five groups (35–39, 40–44, 45–49, 50–54, 55–60). Menopause status was a binary variable (1 = “No”, 2 = “Yes”). In univariate ordinal logistic regression analysis, statistically significant variables were selected and included in multivariate ordinal logistic regression models to explore the relationship between protein-rich food intake, nutrient supplement use, and menopause age and symptoms. Menopausal symptom severity was classified using the modified Kupperman scale as follows: normal (≤6 points), mild (7–15 points), moderate (16–30 points), and severe (>30 points). Specific symptoms were categorized into four dimensions: vasomotor symptoms (hot flashes and sweating), somatic symptoms (palpitations, dizziness, abnormal sensations, skin paresthesia, joint and muscle pain), psychological symptoms (fatigue, irritability, depression, insomnia), and urogenital symptoms (dyspareunia, urinary system diseases) [17,18]. Multivariate linear regression models were used to analyze the relationship between protein-rich foods and nutrient supplements and the four different symptom dimensions. In the multivariate ordinal logistic regression analysis, additionally, subgroup analyses by age group, menopause status, and BMI classification were performed. A two-sided test was used, with a significance level of α = 0.05.

OR_(crude)_ represents the unadjusted odds ratio, indicating the association between protein-rich foods, nutritional supplements, and menopause age without adjusting for potential confounders. OR_(adjust)_ and β-values were calculated after adjusting for age, BMI, menopause status, marital status, smoking and drinking history, education level (≤6 years, 6–12 years, and ≥12 years), exercise habits, sleep quality, frequency of vegetable and fruit intake, sugar-sweetened beverage consumption, and disease history (Having a disease history is defined as the occurrence of any of the following conditions: endometriosis, adenomyosis, fibroids, hypertension (excluding pregnancy-induced hypertension), coronary heart disease, stroke, diabetes, fractures, mumps, depression, anxiety, pelvic floor diseases, pelvic inflammatory disease/chronic pelvic pain, hyperthyroidism, hypothyroidism, malignancies, and those who had received radiation or chemotherapy).

## 3. Results

### 3.1. General Demographic Characteristics

A total of 18,528 participants experienced natural menopause, resulting in a natural menopause rate of 97.60%. The average age of natural menopause was 49.46 ± 3.22 years, with a median menopause age of 50 years. Among the 23,853 participants with a Kupperman score greater than 6, 16,229 had mild symptoms, 7131 had moderate symptoms, and 493 had severe symptoms (Table 1).

### 3.2. Relationship Between Protein-Rich Foods and Nutrient Supplements and Natural Menopause Age

Univariate regression analysis of the impact of protein and nutrient supplements on natural menopause age showed that, compared to those who “never consume”, the consumption of fish, milk, eggs, soy products, vitamin supplements, and calcium supplements was associated with delayed natural menopause. The results of multivariate ordinal logistic regression analysis indicated that, compared to “never consume” participants, fish intake was significantly positively correlated with natural menopause age (Table 2). As the frequency of fish consumption increased, the probability of a delay in natural menopause significantly increased (OR values gradually increased), as shown in Figure 2.

### 3.3. Relationship Between Protein-Rich Foods and Nutrient Supplements and Menopausal Symptoms

The results of univariate regression showed that the consumption of fish, milk, eggs, and soy products was significantly negatively correlated with the severity of menopausal symptoms. Additionally, the use of nutrient supplements was significantly positively correlated with menopausal symptoms. Multivariate regression analysis (Table 3; Figure 3) revealed that the intake of fish, milk, and soy products was significantly negatively correlated with the severity of menopausal symptoms (OR < 1, *p* < 0.001). Furthermore, as the frequency of fish and soy product consumption increased, the probability of alleviating the severity of menopausal symptoms also increased (OR values gradually decreased), as shown in Figure 2. No significant correlation was observed between egg consumption and menopausal symptoms. Moreover, individuals who used calcium and iron nutrient supplements had a significantly higher probability of experiencing worsened menopausal symptoms compared to those who did not use these supplements, with an increase of 48% (OR_adjust_ = 1.48, *p* < 0.001) and 35% (OR_adjust_ = 1.35, *p* < 0.001), respectively.

### 3.4. Relationship Between Protein-Rich Foods and Nutrient Supplements and Different Dimensions of Menopausal Symptoms

Multivariate linear regression analysis results (Table 4) showed that the intake of fish was significantly negatively correlated with somatic symptoms, psychological symptoms, and urogenital symptoms (β < 0, *p* < 0.05). As the frequency of fish consumption increased, the negative impact on the severity of somatic and psychological symptoms also intensified (β values decreased linearly). The intake of milk showed a significant negative correlation with all the dimensions of menopausal symptoms. As the frequency of soy product consumption increased, the negative impact on the severity of the different dimensions of menopausal symptoms also gradually strengthened (β values decreased linearly). The effect of eggs on the different dimensions of menopausal symptoms was not significant. From the β values, it can be observed that the effects of these three types of protein foods on somatic and psychological symptoms were the most pronounced, while the effects on urogenital symptoms and vasomotor symptoms were relatively weaker. Calcium and iron supplements were significantly positively correlated with all the dimensions of menopausal symptoms (*p* < 0.05), whereas vitamin supplements were significantly negatively correlated with urogenital symptoms (*p* < 0.05).

### 3.5. Relationship Between Protein-Rich Foods and Nutrient Supplements and Menopausal Symptoms Across Different Age Groups

Multivariate logistic regression analysis results (Table 5) indicated that, across all age groups, the intake of fish, milk, and soy products was significantly negatively correlated with the severity of menopausal symptoms (OR < 1; *p* < 0.05). In the 35–60 age group, as the frequency of fish and soy product consumption increased, the severity of menopausal symptoms showed a gradual improvement (OR values exhibited a linear decreasing trend). No significant correlation was observed between egg consumption and menopausal symptoms across any age group. The use of calcium and iron supplements was significantly positively correlated with the severity of menopausal symptoms (*p* < 0.05).

### 3.6. Relationship Between Protein-Rich Foods and Nutrient Supplements and Menopausal Symptoms Under Different Menopausal Statuses

Multivariate logistic regression analysis (Table 6) demonstrated that the intake of fish, milk, and soy products was significantly negatively associated with the severity of menopausal symptoms across different menopausal statuses (*p* < 0.001). As the frequency of fish and soy product consumption increased, the severity of menopausal symptoms gradually decreased, showing a linear downward trend in OR values. No significant correlation was observed between egg consumption and menopausal symptoms under the different menopausal statuses. Conversely, the use of calcium and iron supplements was significantly positively correlated with the severity of menopausal symptoms across all the menopausal statuses (*p* < 0.05).

### 3.7. Relationship Between Protein-Rich Foods and Nutrient Supplements and Menopausal Symptoms Across Different BMI Categories

Multivariate logistic regression analysis (Table 7) revealed that among individuals with BMI ≥ 18.5, fish consumption was significantly negatively associated with the severity of menopausal symptoms (OR < 1, *p* < 0.05). In individuals with BMI between 18.5 and 27.9, milk intake was significantly negatively associated with the severity of menopausal symptoms (OR < 1, *p* < 0.05); however, no significant correlation was observed between milk consumption and menopausal symptoms with BMI < 18.5. Additionally, in the normal weight category, the likelihood of alleviated menopausal symptoms increased with higher milk intake frequency, exhibiting a linear downward trend in OR values. Soy product consumption was significantly negatively associated with the severity of menopausal symptoms across all the BMI categories (*p* < 0.05). As the frequency of soy product intake increased, the likelihood of reduced menopausal symptom severity also increased, showing a linear decrease in OR values. In contrast, calcium and iron supplement use was significantly positively associated with the severity of menopausal symptoms in all the BMI categories (OR > 1, *p* < 0.05).

## 4. Discussion

### 4.1. Association of Fish Consumption with Menopause Age and Its Symptoms

The findings of this study demonstrate that higher frequencies of fish consumption are associated with a delay in menopause. This may be attributed to the abundance of omega-3 fatty acids and vitamin D in fish, which play crucial protective roles in women’s reproductive health. Previous cohort studies have shown that fish consumption can effectively delay the onset of natural menopause [19]. Our results further support this conclusion, highlighting a dose–response relationship between fish intake and age of menopause. Specifically, the probability of a delay in menopause increases with higher fish consumption. These findings suggest that increasing fish intake may serve as a simple and effective dietary intervention to help women maintain reproductive health and extend their reproductive lifespan.

In this study, fish consumption was found to be significantly negatively associated with the severity of menopausal symptoms. Previous studies have indicated that fish intake can effectively alleviate menopausal symptoms [20,21,22]. Analysis of the specific symptom dimensions revealed that fish consumption significantly alleviated somatic, psychological, and urogenital symptoms, while no statistically significant relief was observed for vasomotor symptoms. Notably, the negative correlation between fish intake and somatic and psychological symptoms was particularly strong. The subgroup analyses showed that fish consumption was significantly negatively associated with menopausal symptoms across the different age groups, menopausal statuses, and among individuals with a BMI ≥ 18.5. However, this association was not observed in underweight individuals. Although the odds ratio (OR) suggested a protective effect of fish consumption in the underweight population, the results did not reach statistical significance, potentially due to limitations in the data quality and sample size. The potential mechanisms underlying the beneficial effects of fish consumption on menopausal symptoms may include the following. (1) Regulation of neurotransmitters: studies have shown that omega-3 fatty acids in fish promote brain health by enhancing the fluidity of neuronal cell membranes and increasing levels of serotonin and dopamine. These neurotransmitters play a critical role in mood regulation, potentially reducing anxiety and depressive symptoms [23,24]. (2) Anti-inflammatory effects: research suggests that EPA and DHA, two key omega-3 fatty acids, alleviate inflammation by inhibiting nuclear factor kappa-light-chain-enhancer of activated B cells (NF-κB) activation via a peroxisome proliferator-activated receptor gamma (PPAR-γ)-dependent pathway. This anti-inflammatory effect may reduce menopausal symptoms such as joint pain [25]. (3) Improved circulation and cardiovascular health: omega-3 fatty acids have been shown to positively influence cardiovascular health by reprogramming triglyceride-rich lipoprotein metabolism, reducing inflammatory mediators (cytokines and leukotrienes), and regulating cellular adhesion molecules. These effects may alleviate cardiovascular-related menopausal symptoms and reduce hypertension [26,27]. (4) Reduction of hot flashes: omega-3 fatty acids may mitigate hot flashes by regulating thermoregulation and autonomic nervous system balance, which involves serotonin 5-HT43 receptors and norepinephrine pathways in the hypothalamus [28,29,30]. Although the OR values for fish consumption and vasomotor symptoms were less than 1, suggesting a potential protective effect, statistical significance was not reached (*p* > 0.05). Therefore, no definitive conclusion can be drawn. Future studies with larger sample sizes and alternative statistical methods are needed to validate this hypothesis. Overall, this study indicated a dose–response relationship between fish consumption and both delayed age at menopause and alleviation of menopausal symptoms. These findings highlight the importance of adequate fish intake for women’s reproductive health. In particular, women experiencing pronounced somatic and psychological menopausal symptoms may benefit from increased fish consumption.

### 4.2. Association of Milk Consumption with Menopause Age and Its Symptoms

In this study, the univariate logistic regression results suggested that increased milk consumption frequency was associated with a delay in menopause. However, this association was not observed in the multivariate regression analysis. Previous studies have reported that consumption of low-fat or skim milk may delay natural menopause in women under the age of 51, potentially due to increased circulating estrogen levels associated with milk consumption, which could delay the onset of menopause [16,31]. The discrepancy between previous findings and the results of this study highlights the need for more detailed investigations to explore the relationship between dairy consumption and age at natural menopause. Future research should consider dietary patterns, specific types of dairy products, and potential confounding factors to better understand this association.

In the analysis of menopausal symptoms, milk intake was found to be significantly negatively correlated with the severity of menopausal symptoms, which is consistent with findings from previous studies [32,33]. In the analysis of different dimensions of menopausal symptoms, milk consumption was shown to alleviate the severity of various symptoms, including vasomotor symptoms, somatic symptoms, psychological symptoms, and urogenital symptoms. Similar to fish consumption, milk had a more pronounced effect on somatic and psychological symptoms. In the subgroup analyses, milk intake significantly alleviated menopausal symptoms in women of different age groups, menopausal statuses, and those with a BMI of 18.5 ≤ BMI ≤ 27.9. However, this effect was not observed in women with low body weight or obesity. The potential mechanisms underlying the alleviating effects of milk on menopausal symptoms can be explained by the following factors. (1) Vitamin D and calcium: milk is rich in vitamin D and calcium, which are crucial for the health of menopausal women, particularly in alleviating osteoporosis and improving bone mineral density. During the perimenopausal period, the decline in estrogen levels accelerates bone loss; however, vitamin D and calcium enhance bone density, thus reducing the risk of fractures [34]. Vitamin D and calcium not only have significant effects on improving bone density but also play an important role in ovarian protection. Laboratory studies have shown that the ovary is a target organ for 1,25-dihydroxyvitamin D3, and the expression of vitamin D receptors in the ovarian reproductive tissue is significant. Plasma levels of 1,25-dihydroxyvitamin D3 are positively correlated with ovarian reserve, further suggesting the protective role of vitamin D in ovarian aging [35,36]. (2) Cognitive function improvement: studies have shown that the intake of whey protein isolates, which are rich in α-lactalbumin, can improve cognitive performance in stress-sensitive populations, thereby enhancing brain function and reducing the incidence of depression and anxiety. However, the mechanism of dairy products in reducing anxiety and depression is still unclear and requires further investigation [37]. This study found a clear dose–response relationship for milk consumption in women of normal weight, with a more pronounced effect on somatic and psychological symptoms. These findings suggest that perimenopausal women, especially those experiencing significant somatic and psychological symptoms, may benefit from increasing their milk intake.

### 4.3. Association of Soy Products with Menopause Age and Its Symptoms

In this study, the univariate logistic regression results indicated that with an increase in the frequency of soy product consumption, there was a trend toward a delay in natural menopause. In the multivariate logistic regression model, the odds ratio (OR) for soy products in relation to menopausal age was greater than 1, suggesting that soy products may have a certain effect on delaying the onset of natural menopause. However, the statistical analysis did not reach significance, and therefore no definitive conclusion can be drawn. This result may be influenced by the sample size and data conditions. Previous research has suggested that the consumption of soy products may be associated with an earlier onset of menopause [38], while other studies have found no significant impact of soy on menopausal age [39]. There remains controversy regarding the relationship between soy products and the age of menopause, and further research is needed to validate this hypothesis, including increasing sample sizes or conducting longitudinal studies.

In the analysis of menopausal symptoms, the intake of soy products was found to be significantly negatively correlated with the severity of menopausal symptoms, which is consistent with findings from other studies [40,41]. In the analysis of the different dimensions of menopausal symptoms, it was observed that soy products had a relieving effect on symptoms across all the dimensions. The subgroup analysis revealed that in different age groups, menopausal statuses, and BMI categories, soy intake was significantly associated with a reduction in the severity of menopausal symptoms. The mechanisms by which soy products alleviate menopausal symptoms may be explained by several factors. (1) Phytoestrogens: previous randomized controlled trials have shown that soy products are rich in soy isoflavones, which can effectively alleviate menopausal symptoms in women. These compounds are believed to exhibit both estrogenic and anti-estrogenic effects. Isoflavones, which are structurally similar to estradiol, bind with high affinity to β-estrogen receptors. When estrogen levels are low (as in postmenopausal women), isoflavones exhibit estrogen-like effects, whereas they demonstrate anti-estrogenic effects when estrogen levels are high [41]. (2) Antioxidant effects: soy products contain a rich array of antioxidants, including isoflavones, vitamin E, polyphenols, and selenium, which can reduce oxidative stress, protect cell health, and improve overall health. These properties help alleviate discomfort during menopause [42,43]. (3) Bone density maintenance: research has shown that isoflavones reduce the incidence of osteoporosis and play an important role in maintaining bone density [44,45]. This study found a clear dose–response relationship between the intake of soy products and the severity of menopausal symptoms. As the intake of soy products increased, the probability of symptom relief also gradually increased, with a stronger negative correlation with menopausal symptoms across various dimensions. The effect was most pronounced for physical and psychological symptoms. These findings suggest that perimenopausal women may benefit from increasing their intake of soy products to improve menopausal symptoms and promote reproductive health.

### 4.4. Association of Eggs with Menopause Age and Its Symptoms

In this study, the univariate logistic regression results indicated that with an increase in egg consumption frequency, there was a trend toward a delay in natural menopause. The multivariate logistic regression analysis showed that the odds ratio (OR) for eggs and menopausal age was greater than 1, suggesting that egg consumption may have some effect in delaying natural menopause. However, the statistical analysis did not reach significance, and thus no definitive conclusion can be drawn. Some studies have suggested that consuming fresh eggs (≥4 days per week) is associated with an earlier natural menopause [46]. The impact of eggs on female reproductive health warrants further specialized research to verify these findings.

In the analysis of menopausal symptoms, no statistically significant relationship was observed between egg consumption and the severity of menopausal symptoms. Similarly, no statistically significant effects were found for the different dimensions of menopausal symptoms. The subgroup analyses also yielded the same results. Currently, no studies have reported on the relationship between egg consumption and menopausal symptoms. Egg whites are rich in high-quality protein, and the ω-3 fatty acids found in eggs may effectively alleviate vasomotor symptoms of menopause [28,29,30]. However, egg yolks are also rich in cholesterol, and numerous studies have demonstrated that excessive cholesterol intake can adversely affect cardiovascular health. Experimental studies have shown that long-term, high egg consumption (2–4 eggs per day) may lead to higher ratios of LDL-C to HDL-C and elevated LDL-C levels, which may increase the risk of chronic diseases in women [47,48,49,50]. In this study, the subgroup analysis revealed an OR value greater than 1 for eggs and menopausal symptoms, with a positive correlation observed between egg consumption and the severity of physical and psychological symptoms. This suggests that egg consumption may be associated with exacerbating menopausal symptoms. However, due to potential confounding factors, no statistically significant conclusion was drawn. Combining these findings with previous research on the relationship between egg consumption and other chronic diseases suggests that eggs may have a potential exacerbating effect on menopausal symptoms. Therefore, it is recommended that women avoid excessive egg consumption, and further specialized research is needed to validate the relationship between egg intake and menopausal symptoms.

### 4.5. Association of Nutrient Supplements with Menopause Age and Its Symptoms

In this study, the univariate logistic regression results showed that vitamin and calcium supplements were significantly associated with a delay in menopausal age. However, no statistically significant association was observed in the multivariate regression model, which may be due to confounding factors, data conditions, or multicollinearity.

In the analysis of menopausal symptoms, nutrient supplements were found to have a positive impact on the severity of menopausal symptoms, which contradicts the conclusions of previous studies. The analysis of the influencing factors of menopause symptoms from the different dimensions shows that iron supplements have a significant impact on somatic symptoms and vasomotor symptoms. Some studies have indicated a correlation between vasomotor symptoms, osteoarthritis, muscle pain, sensory disturbances and inflammation [51,52,53,54]. These findings suggest that inflammation may play an important role in the occurrence of various menopause symptoms. The possible mechanisms by which iron may exacerbate menopause symptoms can be explained by two factors. First, iron ions can promote the production of reactive oxygen species (ROS), thereby triggering oxidative stress and inflammatory responses. Some research suggests that although iron is an essential element for life, when iron levels in the body are excessive, iron overload, under oxygen-rich conditions, can adversely affect health through oxidative stress. This potentially toxic iron-catalyzed oxidative stress reaction is inevitable [55]. Iron can directly or indirectly (via the Fenton reaction) contribute to the generation of reactive oxygen species (ROS) [56], among which superoxide anion radicals, hydroxyl/free lipid peroxyl radicals, and hydrogen/free lipid peroxyl radicals play significant roles in cellular toxicity [57,58]. Excessive ROS not only directly damages cells but also activates various inflammation-related signaling pathways, thereby exacerbating inflammatory responses [59]. Another possible reason is that individuals who experience menopause symptoms may be more likely to use dietary supplements in an attempt to alleviate these symptoms. Furthermore, after analyzing the data structure, it was found that the incidence of menopausal symptoms was significantly higher in the group that used nutrient supplements. Since this study was a cross-sectional survey, it is unable to determine the temporal relationship between the onset of menopausal symptoms and the use of nutrient supplements. Therefore, further detailed follow-up studies are needed to clarify the impact of nutrient supplements on menopausal symptoms.

## 5. Strengths and Limitations

This study has several strengths, including a large sample size and representative regional selection, with data predominantly collected under the guidance of healthcare professionals, ensuring high authenticity and reliability. Additionally, the scales used in this study have been validated for both reliability and validity, ensuring the scientific rigor of the measurement tools. The innovation of this study lies in its in-depth analysis of the relationship between common protein-rich foods and menopause symptoms as well as age at menopause. Additionally, it provides a preliminary exploration of the potential linear associations between fish and soy products and both age at menopause and menopause symptoms, an area that has been less explored in previous research. Furthermore, this study systematically investigated the relationship between protein-rich foods and various dimensions of menopause symptoms, offering more precise dietary recommendations for women experiencing different types of menopause symptoms. The study also conducted subgroup analyses to examine the relationship between protein-rich foods and menopause symptoms as well as age at menopause across different age groups, menopause statuses, and BMI categories. However, there were several limitations in this study: First, measurement of menopausal age. Although most of the data were collected under the guidance of healthcare professionals, menopausal age was based on self-report, which may be subject to recall bias. However, previous studies have shown that self-reported menopausal age is generally accurate [60]. Second, there were limitations in dietary data. This study had limitations in its dietary data, as it primarily focused on protein-rich foods without considering carbohydrates, meats, or other food categories. Furthermore, specific details on the quantities and proportions of foods consumed were not recorded. This could influence the interpretation of the results. Future research should broaden the scope of dietary assessments, incorporating quantitative data to better understand the relationship between diet and menopausal indicators. Third, there were uncontrolled confounding factors. This study did not account for factors such as hormone therapy use and adverse pregnancy outcomes, which may influence menopausal symptoms and age at menopause, potentially confounding the results. Moreover, lifestyle factors, BMI, and menopausal symptoms can change over time. This study was cross-sectional in design, making it difficult to establish temporal relationships, and therefore, accurate causal inferences cannot be drawn. Future studies should adopt longitudinal designs to dynamically monitor changes in these variables, improving the ability to infer causal relationships. Fourth, there was potential bias in data sources. While some physiological indicators were measured by healthcare professionals, much of the data, including menopausal symptoms, medical history, and general health information, were self-reported by participants, which may have led to information bias. Future research should incorporate objective measures, such as blood biomarkers or imaging assessments, to enhance the accuracy of the data. Fifth, the study may have selection bias. In this study, we used an electronic questionnaire, which ensured the reliability of data management. However, it may also have led to some selection bias, as some participants might have encountered difficulties using the electronic survey. To address this issue, we arranged trained medical staff at all the survey sites to provide face-to-face guidance to the participants, minimizing this selection bias. In summary, this study provides valuable evidence regarding the relationship between dietary factors, natural menopause age, and menopausal symptoms. However, further research using more comprehensive data and longitudinal designs is needed to verify these findings and explore the underlying biological mechanisms.

## 6. Conclusions

The analysis of factors influencing menopausal age revealed a significant association between fish consumption and later age at menopause. Moreover, as the frequency of fish intake increased, menopausal age was progressively delayed, indicating a dose–response relationship. These findings suggest that moderate increases in fish consumption may play a role in delaying natural menopause. In the analysis of factors affecting menopausal symptoms, dietary factors were significantly associated with symptom severity. Specifically, the intake of fish, dairy products, and soy products was significantly negatively correlated with the severity of menopausal symptoms, indicating that these dietary factors may play a positive role in alleviating symptoms, particularly in improving physical and psychological symptoms. However, egg consumption showed no statistically significant effect on the different dimensions of menopausal symptoms. Excessive iron supplementation may be positively correlated with menopause symptoms. The results of this study suggest that increasing the intake of fish, dairy products, and soy products may be an effective strategy for alleviating menopausal symptoms, providing valuable guidance for dietary interventions in perimenopausal women. The use of iron supplements should be approached with caution. However, as this study is cross-sectional in design, causal relationships cannot be established. Future studies should adopt longitudinal designs to further verify the causal associations between dietary factors, natural menopausal age, and menopausal symptoms and to explore the underlying biological mechanisms. In addition, it is recommended to develop personalized dietary intervention programs based on the dietary habits and health conditions of different populations to optimize health management for perimenopausal women.

## Figures and Tables

**Figure 1 nutrients-17-00356-f001:**
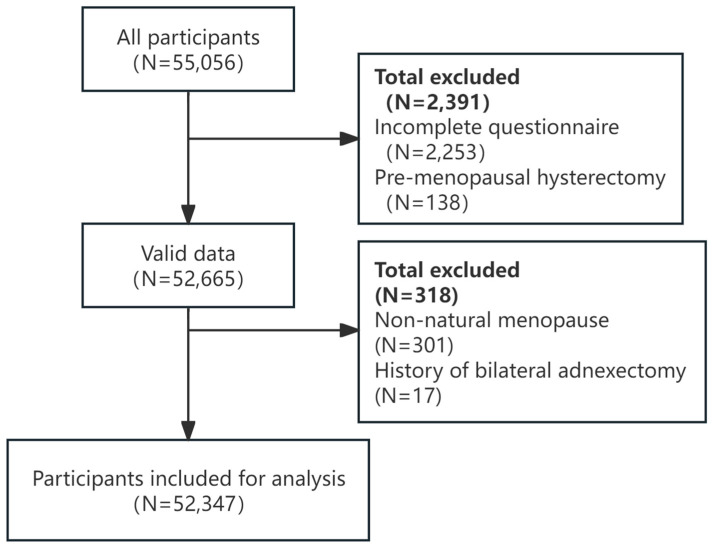
Flowchart of the participants.

**Figure 2 nutrients-17-00356-f002:**
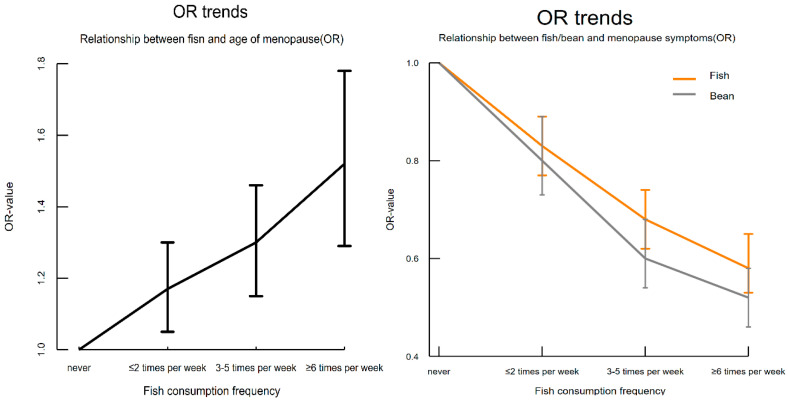
Trend chart of odds ratios (OR).

**Figure 3 nutrients-17-00356-f003:**
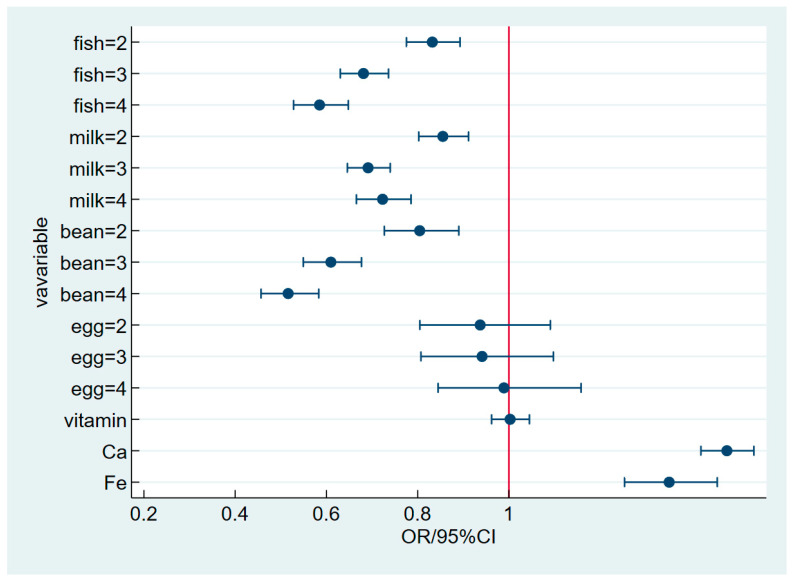
Trend chart of logistics.

**Table 1 nutrients-17-00356-t001:** General demographic characteristics.

Variable		Total No.	Percentage%
Overall		52,347	
Age stage	35–39	10,356	19.78%
	40–44	9340	17.84%
	45–49	10,389	19.85%
	50–54	11,840	22.62%
	55–60	10,422	19.91%
Menopause	Yes	18,528	35.39%
	No	33,819	64.61%
Age of menopause	<40	142	0.77%
	40–44	1130	6.10%
	45–49	7249	39.12%
	50–54	9186	49.58%
	55–60	821	4.43%
	Total	18,528	100%
Kupperman	≤6	28,494	54.43%
	7–15	16,229	31.00%
	16–30	7131	13.62%
	>30	493	0.95%
BMI	<18.5	2003	3.83%
	18.5–23.9	30,252	57.79%
	24.0–27.9	16,301	31.14%
	≥28.0	3791	7.24%
Marriage	Married	49,951	95.42%
	Divorced	1514	2.89%
	Widowed	480	0.92%
	Single	402	0.77%
Education	≤9 years	26,422	50.47%
	9–12 years	10,178	19.44%
	>12 years	15,747	30.09%

**Table 2 nutrients-17-00356-t002:** Association of protein-rich foods and nutrient supplements with menopause age.

	OR_(crude)_	95%CI	*p*-Value	OR_(adjust)_	95%CI	*p*-Value
Fish						
2	1.26	1.15–1.39	0.00	1.17	1.05–1.30	0.00
3	1.47	1.33–1.63	0.00	1.30	1.15–1.46	0.00
4	1.60	1.40–1.83	0.00	1.52	1.29–1.78	0.00
Milk						
2	1.11	1.01–1.21	0.03	0.97	0.87–1.07	0.53
3	1.22	1.11–1.34	0.00	0.97	0.87–1.09	0.63
4	1.13	1.02- 1.26	0.02	0.92	0.81–1.06	0.25
Egg						
2	1.17	0.96–1.42	0.13	1.00	0.79–1.27	0.96
3	1.28	1.06–1.60	0.01	1.04	0.82–1.32	0.77
4	1.26	1.03–1.54	0.02	1.04	0.81–1.33	0.75
Bean						
2	1.25	1.08–1.44	0.00	1.13	0.96–1.32	0.15
3	1.35	1.17–1.56	0.00	1.14	0.97–1.35	0.12
4	1.23	1.04–1.44	0.01	1.02	0.85–1.24	0.81
Vitamin	1.07	1.00–1.15	0.04	1.02	0.95–1.10	0.59
Ca	1.08	1.01–1.14	0.01	1.04	0.98–1.11	0.23
Fe	1.07	0.94–1.22	0.28			

OR_(crude)_ represents the unadjusted odds ratio, indicating the association between protein-rich foods, nutritional supplements, and menopause age without adjusting for potential confounders. OR_(adjust)_ is calculated after controlling for age, BMI, menopause status, marital status, smoking and drinking history, education level, exercise habits, sleep quality, frequency of vegetable and fruit intake, sugar-sweetened beverage consumption, and disease history.

**Table 3 nutrients-17-00356-t003:** Association of protein-rich foods and nutrient supplements with menopausal symptoms.

	OR_(crude)_	95%CI	*p*-Value	OR_(adjust)_	95%CI	*p*-Value
Fish						
2	0.99	0.93–1.06	0.81	0.83	0.77–0.89	0.00
3	0.78	0.73–0.83	0.00	0.68	0.62–0.74	0.00
4	0.64	0.59–0.70	0.00	0.58	0.53–0.65	0.00
Milk						
2	0.90	0.85–0.95	0.00	0.85	0.80–0.91	0.00
3	0.66	0.62–0.70	0.00	0.69	0.65–0.74	0.00
4	0.68	0.64–0.73	0.00	0.72	0.66–0.78	0.00
Egg						
2	0.95	0.84–1.07	0.38	0.94	0.80–1.10	0.40
3	0.81	0.72–0.91	0.00	0.94	0.81–1.10	0.44
4	0.85	0.75–0.96	0.01	0.99	0.84–1.16	0.89
Bean						
2	0.84	0.77–0.91	0.00	0.80	0.73–0.89	0.00
3	0.55	0.51–0.61	0.00	0.60	0.54–0.68	0.00
4	0.47	0.42–0.52	0.00	0.52	0.46–0.58	0.00
Vitamin	1.14	1.09–1.18	0.00	1.00	0.96–1.04	0.93
Ca	1.61	1.55–1.67	0.00	1.48	1.43–1.54	0.00
Fe	1.41	1.31–1.51	0.00	1.35	1.25–1.45	0.00

OR_(crude)_ represents the unadjusted odds ratio, indicating the association between protein-rich foods, nutritional supplements, and menopause symptoms without adjusting for potential confounders. OR_(adjust)_ is calculated after controlling for age, BMI, menopause status, marital status, smoking and drinking history, education level, exercise habits, sleep quality, frequency of vegetable and fruit intake, sugar-sweetened beverage consumption, and disease history.

**Table 4 nutrients-17-00356-t004:** Association of protein-rich foods and nutrient supplements with different dimensions of menopausal symptoms.

	Vasomotor			Somatic			Psychological			Urogenital		
	β	95%CI	*p*-Value	β	95%CI	*p*-Value	β	95%CI	*p*-Value	β	95%CI	*p*-Value
Fish												
2	0.01	−0.06–0.08	0.87	−0.32	−0.41–−0.23	0.00	−0.30	−0.40–−0.21	0.00	−0.06	−0.12–−0.01	0.00
3	−0.03	−0.11–0.05	0.52	−0.59	−0.69–−0.49	0.00	−0.59	−0.69–−0.48	0.00	−0.20	−0.27–−0.13	0.00
4	−0.10	−0.20–0.01	0.07	−0.83	−0.96–−0.70	0.00	−0.83	−0.97–−0.69	0.00	−0.10	−0.18–−0.01	0.02
Milk												
2	−0.06	−0.13–0.01	0.08	−0.28	−0.37–−0.20	0.00	−0.28	−0.37–−0.20	0.00	−0.09	−0.15–−0.04	0.00
3	−0.16	−0.23–−0.09	0.00	−0.56	−0.64–−0.47	0.00	−0.58	−0.67–−0.48	0.00	−0.12	−0.18–−0.06	0.00
4	−0.16	−0.25–−0.07	0.00	−0.49	−0.60–−0.38	0.00	−0.47	−0.59–−0.36	0.00	−0.05	−0.12–0.02	0.17
Egg												
2	−0.11	−0.26–0.05	0.18	0.02	−0.18–0.22	0.84	0.12	−0.08–0.33	0.24	−0.09	−0.22–0.04	0.15
3	−0.21	−0.37–−0.05	0.00	−0.01	−0.21–0.19	0.94	0.19	−0.02–0.40	0.08	−0.07	−0.20–0.06	0.28
4	−0.13	−0.30–0.03	0.11	0.01	−0.19–0.22	0.91	0.30	0.09–0.52	0.01	−0.05	−0.19–0.08	0.43
Bean												
2	−0.21	−0.32–−0.11	0.00	−0.52	−0.66–−0.39	0.00	−0.29	−0.43–−0.15	0.00	−0.16	−0.25–−0.07	0.00
3	−0.33	−0.44–−0.22	0.00	−0.81	−0.95–−0.68	0.00	−0.73	−0.88–−0.59	0.00	−0.33	−0.42–−0.24	0.00
4	−0.35	−0.48–−0.22	0.00	−0.96	−1.12–−0.80	0.00	−1.07	−1.24–−0.91	0.00	−0.30	−0.41–−0.20	0.00
Vitamin	0.04	−0.01–0.08	0.07	−0.01	−0.06–0.05	0.92	0.06	0.01–−0.12	0.03	−0.10	−0.13–−0.06	0.00
Ca	0.30	0.26–0.34	0.00	0.48	0.43–0.53	0.00	0.44	0.39–0.49	0.00	0.20	0.17–0.24	0.00
Fe	0.38	0.30–0.46	0.00	0.53	0.02–0.03	0.00	0.25	0.15–0.36	0.00	0.22	0.15–0.28	0.00

β-value is calculated after controlling for age, BMI, menopause status, marital status, smoking and drinking history, education level, exercise habits, sleep quality, frequency of vegetable and fruit intake, sugar-sweetened beverage consumption, and disease history.

**Table 5 nutrients-17-00356-t005:** Association of protein-rich foods and nutrient supplements with menopausal symptoms across different age groups.

	Age Stage														
	35–39			40–44			45–49			50–54			55–60		
	OR	95%CI	*p*-Value	OR	95%CI	*p*-Value	OR	95%CI	*p*-Value	OR	95%CI	*p*-Value	OR	95%CI	*p*-Value
Fish															
2	0.90	0.73–1.11	0.32	0.78	0.65–0.93	0.00	0.76	0.65–0.90	0.00	0.80	0.70–0.92	0.00	0.88	0.77–1.01	0.07
3	0.67	0.54–0.83	0.00	0.59	0.48–0.72	0.00	0.65	0.54–0.77	0.00	0.68	0.58–0.79	0.00	0.79	0.68–0.92	0.00
4	0.55	0.42–0.73	0.00	0.41	0.32–0.54	0.00	0.54	0.43–0.69	0.00	0.62	0.51–0.76	0.00	0.77	0.62–0.95	0.01
Milk															
2	0.79	0.67–0.94	0.01	0.81	0.69–0.94	0.01	0.90	0.78–1.04	0.15	0.87	0.76–0.98	0.03	0.89	0.78–1.02	0.08
3	0.63	0.52–0.75	0.00	0.59	0.50–0.70	0.00	0.74	0.64–0.86	0.00	0.68	0.60–0.78	0.00	0.84	0.73–0.97	0.02
4	0.53	0.43–0.66	0.00	0.63	0.52–0.78	0.00	0.73	0.61–0.88	0.00	0.81	0.68–0.95	0.01	0.94	0.78–1.11	0.45
Egg															
2	0.62	0.04–0.90	0.01	0.81	0.53–1.21	0.30	1.37	0.95–1.97	0.09	0.94	0.70–1.27	0.70	1.04	0.77–1.40	0.80
3	0.71	0.48–1.03	0.07	0.88	0.58–1.33	0.55	1.31	0.91–1.89	0.15	0.92	0.68–1.24	0.58	0.94	0.70–1.28	0.70
4	0.77	0.52–1.14	0.19	0.98	0.64–1.50	0.94	1.44	0.99–2.10	0.06	0.95	0.69–1.30	0.75	0.83	0.61–1.14	0.25
Bean															
2	0.68	0.52–0.87	0.00	0.96	0.75–1.23	0.74	0.73	0.58–0.92	0.01	0.84	0.68–1.03	0.09	0.81	0.66–0.99	0.04
3	0.54	0.41–0.71	0.00	0.75	0.58–0.97	0.03	0.53	0.42–0.67	0.00	0.63	0.51–0.78	0.00	0.60	0.49–0.75	0.00
4	0.51	0.38–0.70	0.00	0.60	0.44–0.82	0.00	0.39	0.29–0.51	0.00	0.48	0.38–0.62	0.00	0.63	0.49–0.81	0.00
Vitamin	1.04	0.94–1.14	0.47	1.04	0.95–1.15	0.35	0.99	0.90–1.09	0.85	0.99	0.90–1.08	0.75	1.00	0.91–1.10	0.99
Ca	1.26	1.13–1.39	0.00	1.33	1.20–1.47	0.00	1.44	1.33–1.57	0.00	1.75	1.43–1.66	0.00	1.62	1.49–1.76	0.00
Fe	1.44	1.22–1.69	0.00	1.43	1.20–1.71	0.00	1.38	1.17–1.63	0.00	1.18	1.10–1.52	0.00	1.38	1.16–1.64	0.00

OR is calculated after controlling for BMI, menopause status, marital status, smoking and drinking history, education level, exercise habits, sleep quality, frequency of vegetable and fruit intake, sugar-sweetened beverage consumption, and disease history.

**Table 6 nutrients-17-00356-t006:** Association of protein-rich foods and nutrient supplements with menopausal symptoms under different menopausal statuses.

	Menopause					
	Yes			No		
	OR	95%CI	*p*-Value	OR	95%CI	*p*-Value
Fish						
2	0.82	0.74–0.91	0.00	0.83	0.76–0.92	0.00
3	0.69	0.62–0.78	0.00	0.67	0.60–0.75	0.00
4	0.66	0.56–0.77	0.00	0.55	0.48–0.63	0.00
Milk						
2	0.86	0.78–0.95	0.00	0.86	0.79–0.94	0.00
3	0.73	0.65–0.80	0.00	0.68	0.62–0.75	0.00
4	0.87	0.76–0.99	0.03	0.66	0.59–0.74	0.00
Egg						
2	0.99	0.79–1.24	0.96	0.91	0.74–1.11	0.36
3	0.94	0.75–1.18	0.60	0.95	0.77–1.17	0.62
4	0.84	0.67–1.07	0.16	1.08	0.88–1.34	0.47
Bean						
2	0.79	0.68–0.92	0.00	0.81	0.71–0.92	0.00
3	0.61	0.52–0.72	0.00	0.61	0.53–0.70	0.00
4	0.54	0.44–0.65	0.00	0.49	0.42–0.58	0.00
Vitamin	1.00	0.93–1.07	0.95	1.01	0.96–1.06	0.66
Ca	1.58	1.48–1.68	0.00	1.39	1.32–1.46	0.00
Fe	1.34	1.18–1.53	0.00	1.38	1.26–1.51	0.00

OR is calculated after controlling for age, BMI, marital status, smoking and drinking history, education level, exercise habits, sleep quality, frequency of vegetable and fruit intake, sugar-sweetened beverage consumption, and disease history.

**Table 7 nutrients-17-00356-t007:** Association of protein-rich foods and nutrient supplements with menopausal symptoms across different BMI categories.

	Underweight			Normal Weight			Overweight			Obesity		
	OR	95%CI	*p*-Value	OR	95%CI	*p*-Value	OR	95%CI	*p*-Value	OR	95%CI	*p*-Value
Fish												
2	1.35	0.93–1.97	0.11	0.86	0.78–0.94	0.00	0.78	0.70–0.88	0.00	0.77	0.61–0.97	0.03
3	1.17	0.78–1.76	0.45	0.68	0.61–0.76	0.00	0.67	0.59–0.76	0.00	0.62	0.47–0.80	0.00
4	0.83	0.49–1.40	0.48	0.59	0.51–0.67	0.00	0.57	0.47–0.69	0.00	0.61	0.42–0.90	0.01
Milk												
2	0.74	0.53–1.04	0.08	0.86	0.79–0.94	0.00	0.83	0.75–0.92	0.00	0.92	0.75–1.14	0.44
3	0.65	0.45–0.93	0.02	0.67	0.61–0.74	0.00	0.70	0.62–0.79	0.00	0.73	0.58–0.92	0.00
4	0.78	0.51–1.19	0.25	0.64	0.58–0.72	0.00	0.82	0.71–0.94	0.01	1.00	0.74–1.36	0.98
Egg												
2	1.59	0.74–3.46	0.24	0.97	0.78–1.21	0.82	0.78	0.61–1.00	0.06	1.22	0.76–1.95	0.41
3	1.27	0.59–2.76	0.54	1.07	0.86–1.32	0.57	0.74	0.57–0.95	0.02	0.99	0.62–1.60	0.97
4	1.32	0.59–2.93	0.50	1.17	0.93–1.46	0.18	0.71	0.55–0.93	0.01	1.05	0.64–1.72	0.84
Bean												
2	0.74	0.43–1.27	0.28	0.84	0.73–0.97	0.02	0.80	0.67–0.94	0.01	0.62	0.44–0.88	0.01
3	0.50	0.29–0.87	0.01	0.62	0.53–0.71	0.00	0.66	0.55–0.78	0.00	0.46	0.32–0.65	0.00
4	0.34	0.18–0.64	0.00	0.49	0.42–0.58	0.00	0.60	0.49–0.74	0.00	0.45	0.29–0.70	0.00
Vitamin	0.81	0.66–1.00	0.05	1.04	0.99–1.10	0.11	0.94	0.87–1.02	0.15	1.06	0.90–1.25	0.49
Ca	1.61	1.31–1.99	0.00	1.43	1.36–1.51	0.00	1.54	1.44–1.66	0.00	1.58	1.37–1.83	0.00
Fe	1.44	1.00–2.05	0.05	1.30	1.18–1.44	0.00	1.38	1.20–1.58	0.00	1.57	1.16–2.12	0.00

OR is calculated after controlling for age, menopause status, marital status, smoking and drinking history, education level, exercise habits, sleep quality, frequency of vegetable and fruit intake, sugar-sweetened beverage consumption, and disease history.

## Data Availability

The original contributions presented in this study are included in the article. Further inquiries can be directed to the corresponding authors.

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
