# Peer review of "Association Between Protein-Rich Foods, Nutritional Supplements, and Age of Natural Menopause and Its Symptoms"

_nutrients, 2025, doi:10.3390/nu17020356_

Round 1
Reviewer 1 Report
Comments and Suggestions for Authors
Association between Protein-Rich Foods, Nutritional Supplements, and Natural Menopause Age Menopausal Symptoms
Title: second part of the title is not clear. Please revise it to make it more understandable, like “..menopausal symptoms of natural-age menopause” or “age of natural menopause and its symptoms”
Overall: The topic is important medically and have a huge value form many patients worldwide. Important information about detrimental effect of iron and calcium supplements is shown, and possible cause-effect relation can be suggested. Unfortunately no data on analyzed factor (except fish consumption) in relation to the age of menopause was shown. Further interventional studies needs to be done to draw clinical implications from observed associations.
Abstract: Generally well written but revisions have to be done:
Information about detrimental effects of iron and calcium should be highlighted!
V 11 – revise the sentence to make it clear as in the title
V 24 I do not understand “menopausal status” in this context – do you assess menopausal symptoms in women without menopause? Menopause have no effect on menopausal symptoms? Please clarify the information, since it is not easy to understand the idea behind.
Keywords: influencing factors and logistic regression are not the relevant keywords – please use fish, milk, soy instead
Introduction :
It will be good to show clear definition of menopause (practice definition that will be used in the study) in the introduction.
Material and methods:
How the last menstrual period was defined for patients – do they obtain the instructions about it? – how they distinguish postmenstrual bleeding from menopause. What was the duration of amenorrhea that was enough to define the “last menstrual bleeding” – please clarify
Results:
Please revise the names of figures and tables – they should be more precise and clear
Discussion:
Authors comprehensively discussed main outcomes obtained, showing possible mechanisms and relation of results to current literature. Unfortunately discussion on iron intake in supplements is missing. It is well known that iron ions can increase the production of Reactive oxygen species ROS and increase inflammation. It will be interesting to show that link in the discussion.
Potential limitations of the study were indicated correctly and precisely.
Conclusions:
Well composed. Further information about iron supplements have to be included after revision of mansucript
Reviewer 2 Report
Comments and Suggestions for Authors
This is a very valuable research due to its design, methodology and sample size, and usefulness for women's public health, and paper is well-structured and written. This study has many strengths and I think that its communication will be important for the scientific community.
However, I have the following comments and suggestions for the authors:
- Abstract: lines 10-11. I suggest the change highlighted to a higher precision:
"To investigate the relationship between protein-rich foods, various nutritional supplements and natural menopause age, AND/OR menopausal symptoms"
- line 30: conclusion: Although the authors recognize in discussion section that their design does not allow causality to be established, in the conclusion of the abstract they suggest it as their main finding. I suggest to change to state their main conclusions :their findings of association, which suggests that may play a role as intervention ... but not to state that may play a role (speculative) without including their conclusion (the finding of association).
- line 36: I do not understand what "the physiological process of women" means talking about their menopause. I suggest to improve the sentence wording to a better comprehension
- lines 74-81: Inclusion an exclusion criteria might be showed through a classic flow-chart. And in lines 78-79 the authors state: "Women with non-natural menopause (301 78 participants) and a history of bilateral adnexectomy (17participants) were also excluded," Actually women with bilateral adnexectomy might be already included among non-natural menopause ...
- line 117: the author distinguish between endometriosis and chocolate cysts and both are endometriosis ...
- Table 1. For a better understanding I suggest to introduce an horizontal line between sections of demographic characteristic distribution, and I think that the total considered for each characteristic might be included (for example n=x) because the total is not the same for each of them.
- Table 2: the title might include that is showed multivariate logistic regression
- the legend of figures and tables might include the complete description of what are showing of including. Applicable for figure 1 and 2
- The titles of tables, and discussion sections state "Effects" ... but I think that might be changed to "Relationship" or similar. Effect suggests causality and it is not tested in this study without intervention. They only studied associations.
- Line 346-7: "Research has shown that isoflavones help maintain bone density and reduce the incidence of osteoporosis, which is crucial for alleviating menopausal symptoms in women"
This is not correct. Osteoporosis is a frequent and important repercussion of menopause but it is an asymptomatic condition if a fracture is not present ... it might be changed.
- I think that reference number 13 is not useful for most of readers of this journal
Other general comments:
- The authors mention many times "delayed menopause" without definition of it and actually referring to "delay in menopause" or "later menopause".
- The authors often claim that analyzed nutrients can benefit or alleviate menopausal symptoms but this is only speculative from their study. The design of the study only afford to conclude the finding of associations or not, and this suggests that a benefit of intake could exist for menopausal women with symptoms for example. But to investigate it, an intervention study would be necessary. So I recommend to review the text being cautious in this sense with the wording.
Reviewer 3 Report
Comments and Suggestions for Authors
Thank you very much for allowing me to review the work entitled “ nutrients-3410955_ Association between Protein-Rich Foods, Nutritional Supplements, and Natural Menopause Age Menopausal Symptoms”, which investigates the relationship between protein-rich foods, various nutritional supplements, and the natural age of menopause, as well as menopausal symptoms.
The title is appropriate to the content of the work and provides relevant information about it.
Regarding the abstract, the methods section describes the sample recruitment procedure but does not identify the study design. As described, it appears to be a cross-sectional design, which should be explicitly stated.
In the introduction, the hypothesis about the potential role of supplements and protein-rich foods in influencing menopause should be expanded upon, providing a stronger scientific basis or rationale for this hypothesis using appropriate references.
In the materials and methods section, as with the abstract, the study design is missing. The use of an electronic questionnaire should be discussed, particularly whether it restricts access for all women meeting the inclusion criteria to complete the information. This should be evaluated, as women with access to the electronic questionnaire may differ systematically from the broader population of women, introducing potential selection bias. Additionally, the self-reported nature of the data could lead to discrepancies in how individual participants evaluate their experiences, introducing information bias. The specific questionnaire used should be identified or attached as a supplementary document.
In the statistical analysis section, it should be stated how the normality of distribution was assessed. Variables with a non-normal distribution should be represented as medians and ranges.
In lines 114–122, different habits and pathologies are mentioned. The rationale for adjusting for these specific habits and pathologies should be included in the introduction and discussed in the discussion section.
For the results, it should be clearly indicated in the tables which values represent crude odds ratios and which represent adjusted odds ratios. This distinction is crucial, as dietary components often interact, and their combined effects must be assessed collectively.
The discussion is highly interesting, as it highlights the plausibility of a dose-response relationship supporting causality. However, given that the study appears to be cross-sectional, only associations can be determined, although these point towards potential causality. The need to confirm these findings with further research employing more robust methodologies and higher levels of scientific evidence should be emphasised, especially considering the promising results of this study.
In terms of limitations, it should be noted that the use of an electronic, self-reported survey by participants is not discussed. This factor must be considered and acknowledged.
Round 2
Reviewer 3 Report
Comments and Suggestions for Authors
Thank you very much for giving me the opportunity to review once again the work titled “nutrients-3410955_ Association between Protein-Rich Foods, Nutritional Supplements, and Natural Menopause Age Menopausal Symptoms”, which analyses the factors influencing menopausal age and reveals a significant association between fish consumption and a later age at menopause.
I sincerely appreciate the effort your have put into improving the manuscript. The clarifications provided regarding the requested points, as well as the inclusion of additional information, have greatly enhanced the clarity and understanding of the study.
Regarding the introduction, the discussion of the controversial nature of the existing literature on this topic strengthens the rationale for conducting this research, as it aims to shed light on this complex issue.